# Class-Distribution-Aware Pseudo-Labeling for Semi-Supervised Multi-Label Learning

**Ming-Kun Xie**[1,2], **Jia-Hao Xiao**[1,2], **Hao-Zhe Liu**[1,2], **Gang Niu**[3], **Masashi Sugiyama**[3,4],
**Sheng-Jun Huang**[1,2*]

[1]Nanjing University of Aeronautics and Astronautics
[2]MIIT Key Laboratory of Pattern Analysis and Machine Intelligence, Nanjing, China
[3]RIKEN Center for Advanced Intelligence Project
[4]The University of Tokyo, Tokyo, Japan
`{mkxie, jiahaoxiao, haozheliu, huangsj}@nuaa.edu.cn`
`gang.niu.ml@gmail.com  sugi@k.u-tokyo.ac.jp`

## Abstract

Pseudo-labeling has emerged as a popular and effective approach for utilizing unlabeled data. However, in the context of semi-supervised multi-label learning (SSMLL), conventional pseudo-labeling methods encounter difficulties when dealing with instances associated with multiple labels and an unknown label count. These limitations often result in the introduction of false positive labels or the neglect of true positive ones. To overcome these challenges, this paper proposes a novel solution called Class-Aware Pseudo-Labeling (CAP) that performs pseudo-labeling in a class-aware manner. The proposed approach introduces a regularized learning framework incorporating class-aware thresholds, which effectively control the assignment of positive and negative pseudo-labels for each class. Notably, even with a small proportion of labeled examples, our observations demonstrate that the estimated class distribution serves as a reliable approximation. Motivated by this finding, we develop a class-distribution-aware thresholding strategy to ensure the alignment of pseudo-label distribution with the true distribution. The correctness of the estimated class distribution is theoretically verified, and a generalization error bound is provided for our proposed method. Extensive experiments on multiple benchmark datasets confirm the efficacy of CAP in addressing the challenges of SSMLL problems. The implementation is available at `https://github.com/milkxie/SSMLL-CAP`.

## 1   Introduction

In *single-label* supervised learning, each instance is assumed to be associated with only one class label, while many realistic scenarios may be *multi-labeled*, where each instance consists of multiple semantics. For example, an image of a nature landscape often contains the objects of *sky*, *cloud*, and *mountain*. Multi-label learning (MLL) is a practical and effective paradigm for handling examples with multiple labels. It trains a classifier that can predict all the relevant labels for unseen instances based on the given training examples. A large number of recent works have witnessed the great progress that MLL has made in many practical applications [27], *e.g.*, image annotation [21], protein subcellular localization [28], and visual attribute recognition [32].

Thanks to its powerful capacity, the deep neural network (DNN) has become a prevalent learning model for handling MLL examples [34]. Unfortunately, it requires a large number of precisely labeled

---

*Correspondence to: Sheng-Jun Huang (huangsj@nuaa.edu.cn).

37th Conference on Neural Information Processing Systems (NeurIPS 2023).

examples to achieve favorable performance. This leads to a high cost of manual annotation, especially when the dataset is large and the labeling task must be carried out by an expert. Given that it is hard to train an effective DNN based on a small subset of training examples, it is rather important to exploit the information from unlabeled instances. The problem has been formalized as a learning framework called semi-supervised multi-label learning (SSMLL), which aims to train a classifier based on a small set of labeled MLL examples and a large set of unlabeled ones.

Compared to semi-supervised learning (SSL) that has made great progress [4, 38], SSMLL has received relatively less attention in the context of deep learning. Generally, there are still three main challenges towards the development of SSMLL. Firstly, since each instance is associated with multiple labels and the number is unknown, the commonly used pseudo-labeling strategy that selects the most probable label or the top-$k$ probable labels cannot be applied to the SSMLL problems. It would face the dilemma of either introducing false positive labels or neglecting true positive ones. Secondly, due to the intrinsic class-imbalance property of MLL data, it is hard to achieve favorable performance by using a fixed threshold for each instance. Thirdly, recent studies have mainly focused on multi-label learning with missing labels (MLML) [11, 17, 2] scenarios, where each training instance is assumed to be assigned with a subset of true labels. Unfortunately, these methods often fail to achieve favorable performance, or cannot even be applied to the SSMLL scenarios, since most of them were designed under the assumption of MLML.

To solve these challenges, in this paper, we propose a novel Class-Aware Pseudo-labeling (CAP) method for handling the SSMLL problems. Unlike the existing methods, we perform pseudo-labeling in a class-aware manner to avoid estimating the number of true labels for each instance, which can be very hard in practice. Specifically, a regularized learning framework is proposed to determine the numbers of positive and negative pseudo-labels for each class based on the class-aware thresholds. Given that the true class distribution is unknown, we alternatively determine the thresholds based on the estimated class distribution of labeled data, which can be a tight approximation according to our observation. Our theoretical results show the correctness of estimated class distribution and provide a generalization error bound for CAP. Extensive experimental results on multiple benchmark datasets with a variety of comparing methods validate that the proposed method can achieve state-of-the-art performance.

## 2 Related Work

Thanks to the powerful learning capacity of DNNs, MLL has made great advances in the context of deep learning. Some methods designed architectures [6] or training strategies [22, 49] to exploit the label correlations. Some other methods designed sophisticated loss functions to improve the performance of MLL [34]. The last group of methods designed specific architectures to capture the objects related to semantic labels. Global-average-pooling (GAP) based models [33] and attention-based models [22, 26] are two groups of representative methods.

There are relatively few works that study how to improve the performance of deep models in SSMLL scenarios. Instead of end-to-end training, the only deep SSMLL method [43] performed the two-stage training, which first used a DNN to extract features, and then used a linear model to perform classification. [37] proposed a deep sequential generative model to handle the noisy labels collected by crowdsourcing and unlabeled data simultaneously. [20] focused on the transductive and non-deep scenario, and thus cannot be applied to our setting. The method proposed by [39] utilized the graph neural network (GNN) to deal with SSMLL data with graph structures. In contrast, there are many works that trained linear models to solve the SSMLL problems [5, 15, 42, 52, 50, 40]. M3DN was proposed to deal with SSMLL data in multi-modal multi-instance learning scenarios by adopting optimal transport technique [48].

Pseudo-labeling has become a popular method in semi-supervised learning (SSL). The idea was firstly applied to semi-supervised training of deep neural networks [23]. Subsequently, a great deal of works have been devoted to improving the quality of pseudo-labels either by adopting consistency regularization [4, 38], or by using distribution alignment [30, 3]. The contrastive learning technique has been applied to improve the performance of SSL [24]. To improve the reliability of pseudo-labels, an uncertainty-aware pseudo-labeling method proposed in [35] selected reliable pseudo-labels based on the prediction uncertainty. Recent studies have also paid attention to dealing with the class-imbalance problem of pseudo-labeling in SSL scenarios [18, 45, 14]. Unlike FixMatch that

selects unlabeled examples with a fixed threshold, Dash [47] selected unlabeled examples with a dynamic threshold, with the goal of achieving better pseudo-labeling performance. FlexMatch [51] was proposed to select unlabeled examples for every class according to the current learning status of the model. Several works have been explored the idea of selecting different thresholds for different classes to improve the performance of SLL [13, 14, 44].However, these methods are designed for the multi-class single-label scenario, and cannot be directly applied to the multi-label scenario.

In order to reduce the annotation cost, a cost-effective strategy is to assign a subset of true labels to each instance. For example, [11] designed a partial binary cross entropy (BCE) loss that re-weights the losses of known labels. As an extreme case of MLML, single positive multi-label learning (SPML) [8, 53, 41] assumes that only one of multiple true labels can be observed during the training stage. The pioneering work [8] trains DNNs by simply treating unobserved labels as negative ones and utilizes the regularization to alleviate the harmfulness of false negative labels. [53] propose asymmetric pseudo labeling technique to recover true labels.

## 3  The Method

In the SSMLL problem, let $x \in \mathcal{X}$ be a feature vector and $y \in \mathcal{Y}$ be its corresponding label vector, where $\mathcal{X} = \mathbb{R}^d$ is the feature space and $\mathcal{Y} = \{0, 1\}^q$ is the label space with $q$ possible class labels. Here, $y_k = 1$ indicates the $k$-th label is relevant to the instance, while $y_k = 0$, otherwise. Suppose that we are given a labeled dataset with $n$ training examples $\mathcal{D}_l = \{(x_i, y_i)\}_{i=1}^n$ and an unlabeled dataset with $m$ training instances $\mathcal{D}_u = \{x_j\}_{j=1}^m$. Our goal is to train a DNN $f(x; \theta)$ based on the labeled dataset $\mathcal{D}_l$ and unlabeled dataset $\mathcal{D}_u$, where $\theta$ is the parameter of the network. For notational simplicity, we omit the notation $\theta$ and let $f(x)$ be the predicted probability distribution over classes and $f_k(x)$ be the predicted probability of the $k$-th class for input $x$.

Typical multi-label learning methods usually train a DNN with the commonly used binary cross entropy (BCE) loss, which decomposes the original task into multiple binary classification problems. Unfortunately, BCE loss often suffers from positive-negative imbalance issue. To mitigate this problem, we adopt the asymmetric loss (ASL) [34], which is a variant of focal loss with different focusing parameters for positive and negative instances. In our experiment, we found it works better than BCE loss. Formally, given the predicted probabilities $f(x)$ on instance $x$, the ASL loss is defined as

$$\mathcal{L}(f(x), y) = \sum_{k=1}^q y_k \ell_1(f_k(x)) + (1 - y_k)\ell_0(f_k(x)), \tag{1}$$

Here, $\ell_1(f_k) = -(1 - f_k)^{\lambda_1} \log(f_k)$ and $\ell_0(f_k) = -(f_k)^{\lambda_0} \log(1 - f_k)$ represent the losses calculated on positive and negative labels, where $\lambda_1$ and $\lambda_0$ are positive and negative focusing parameters.

### 3.1  Instance-Aware Pseudo-Labeling

The loss function may not be the best choice to solve the SSMLL problem, since besides the labeled training examples, there still exist a large number of unlabeled training examples. To exploit the information of unlabeled data, inspired by recent SSL works [4, 38], an intuitive strategy is assigning the unlabeled instances with pseudo-labels based on the model outputs. Formally, we define the unlabeled loss $\mathcal{L}_u$ as

$$\mathcal{L}_u(f(x), \hat{y}) = \sum_{k=1}^q \hat{y}_k \ell_1(f_k(x)) + (1 - \hat{y}_k)\ell_0(f_k(x)),$$

where $\hat{y} = [\hat{y}_1, \cdots, \hat{y}_j]^\top$ represents the pseudo-label vector for instance $x$.

In the above formulation, the most significant element is how to obtain the pseudo-labels $\hat{y}$ that significantly affects the final performance of SSMLL. Most of existing pseudo-labeling methods are performed in an instance-aware manner by assigning pseudo-labels to each unlabeled instance based on its probability distribution. Below, we briefly review three instance-aware pseudo-labeling strategies that can be applied to the SSMLL problems. The most commonly used strategy adopted by the SSL method called FixMatch [38] is to select the most probable label as the ground-truth one:

$$\hat{y}_k = \begin{cases} 1 & \text{if } k = \arg\max_{c \in [q]} f_c(x), \\ 0 & \text{otherwise.} \end{cases} \tag{2}$$

One advantage of the strategy is that it is likely to safely identify a true label for each unlabeled training instance. Unfortunately, it is obvious that the strategy would neglect multiple true labels. Generally, it transforms the unlabeled dataset into another learning scenario called single positive multi-label learning (SPML) [8], where only one of multiple positive labels is available for each instance. A straightforward strategy is to simply treat unobserved labels as negative ones. Although this strategy enables us to train a classifier based on SPML data, it would introduce a large number of false negative labels, leading to unfavorable performance.

The second choice is an improved version of the above strategy, which selects the top $l$ probable labels as the true ones:

$$\hat{y}_k = \begin{cases} 1 & \text{if } f_k(\boldsymbol{x}) \geq \tau^l, \\ 0 & \text{otherwise,} \end{cases} \tag{3}$$

where $\tau^l$ is the $l$-th predicted probability in a descending order. The strategy conducts a competition among labels, and selects top $l$ winners. The optimal solution is to set $l$ as the true number of positive labels for each unlabeled instance. Unfortunately, since the true number is unknown in practice, as a compromise, we set $l$ as the average number of positive labels per instance. Given that the true number does not always equal to the average number, it would be caught in a dilemma of either introducing false positive labels or neglecting true positive ones.

The last choice is to adopt an instance-aware threshold $\tau_j$ that separates positive and negative labels for each unlabeled instance.

$$\hat{y}_{jk} = \begin{cases} 1 & \text{if } f_k(\boldsymbol{x}_j) \geq \tau_j, \\ 0 & \text{otherwise.} \end{cases} \tag{4}$$

Compared to the above methods, this strategy achieves a strong flexibility that allows it to assign different numbers of positive labels to different instances. A potential limitation is that it is hard to find the optimal thresholds for different instances. In practice, a feasible solution is to adopt a global threshold, that is $\forall j \in [m], \tau_j = \tau$. Obviously, it is impossible to adopt a global threshold that is optimal for all instances, especially considering the class-imbalance property of MLL data. In general, a large threshold often leads to a small recall score of tail classes, which indicates that less positive labels would be identified. While a small threshold often results in a small precision score of head classes, which indicates besides positive labels, a great deal of negative ones would be treated as positive ones. The dilemma prevents the model from obtaining favorable performance.

### 3.2 Class-Aware Pseudo-Labeling

As discussed above, in many real-world scenarios, it is really difficult to acquire the true number of positive labels for each instance. This leads the instance-aware pseudo-labeling methods to be caught in the dilemma of either mislabeling false positive labels or neglecting true positive labels, resulting in a noticeable decrease of the model performance.

To solve this issue, we propose a regularized learning framework to assign pseudo-labels in a class-aware manner. Formally, we reformulate the optimization problem of SSMLL as

$$\min_{\hat{\boldsymbol{y}}, \theta} \sum_{i=1}^{n} \sum_{k=1}^{q} y_{ik} \ell_1(f_k(\boldsymbol{x}_i)) + (1 - y_{ik}) \ell_0(f_k(\boldsymbol{x}_i))$$

$$+ \sum_{j=1}^{m} \sum_{k=1}^{q} \hat{y}_{jk} \ell_1(f_k(\boldsymbol{x}_j)) + (1 - \hat{y}_{jk}) \ell_0(f_k(\boldsymbol{x}_j)) - \sum_{j=1}^{m} \sum_{k=1}^{q} \alpha_k \hat{y}_{jk} + \beta_k (1 - \hat{y}_{jk}), \tag{5}$$

$$\text{s.t.} \quad \forall j \in [m], \hat{\boldsymbol{y}}_j = [y_{j1}, \cdots, y_{jq}]^\top \in \{0, 1\}^q,$$

$$\forall k \in [q], \alpha_k > 0, \beta_k > 0,$$

where $\alpha_k$ and $\beta_k$ are class-aware regularized parameters to control how many positive and negative labels would be included into model training for class $k$. Below, we primarily provide a solution of the optimization problem Eq.(5), and then discuss how to set parameters $\alpha_k$ and $\beta_k$ to capture the true class distribution of unlabeled examples.

**Alternative Search**    It is hard to directly solve the optimization problem Eq.(5), since there are two sets of variables. A feasible solution is to adopt the alternative convex search [1, 54] strategy that optimizes a group of variables by fixing the other group of variables.

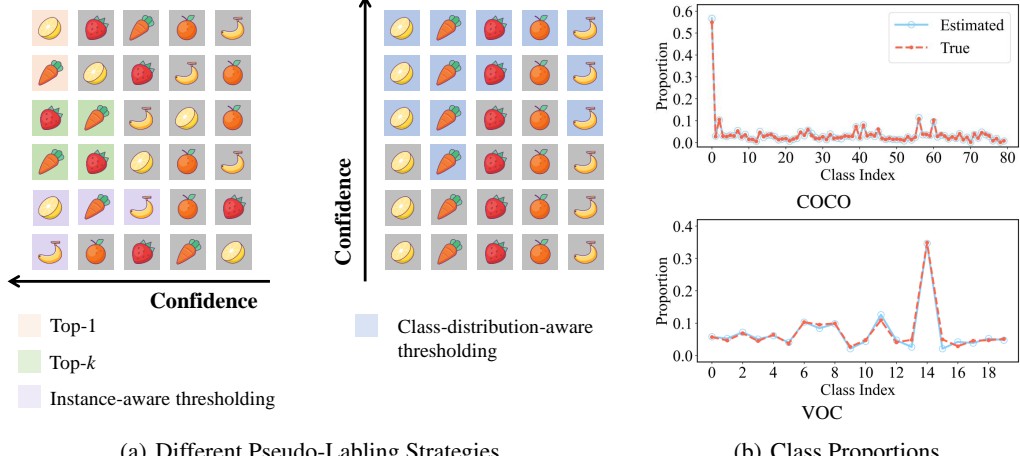

(a) Different Pseudo-Labling Strategies  (b) Class Proportions

Figure 1: (a) An illustration of the comparison between instance-aware and class-aware pseudo-labeling methods. (b) The curves of the estimated and true class proportions on COCO and VOC. By using the CAT strategy, CAP can provide high-quality pseudo-labels by approximating the true class distribution. This can be validate by the results in (b), where the empirical and true class proportions of positive labels show high-level consistency.

Suppose that pseudo labels $\hat{\boldsymbol{y}}$ are given, then the optimization problem Eq.(5) can be transformed into an ordinary loss by treating the pseudo labels as the true ones:

$$\min_{\theta} \frac{1}{n} \sum_{i=1}^{n} \mathcal{L}(f(\boldsymbol{x}_i), \boldsymbol{y}_i) + \frac{1}{m} \sum_{j=1}^{m} \mathcal{L}_u(f(\boldsymbol{x}_j), \hat{\boldsymbol{y}}_j), \tag{6}$$

which can be solved by applying the stochastic gradient decent (SGD) method.

With the parameters $\theta$ fixed, we reformulate the optimization problem with respect to $\hat{\boldsymbol{y}}$ as

$$\min_{\hat{\boldsymbol{y}}} \sum_{j=1}^{m} \sum_{k=1}^{q} \hat{y}_{jk} \ell_1(f_k(\boldsymbol{x}_j)) + (1 - \hat{y}_{jk}) \ell_0(f_k(\boldsymbol{x}_j)) - \sum_{j=1}^{m} \sum_{k=1}^{q} \alpha_k \hat{y}_{jk} + \beta_k(1 - \hat{y}_{jk}). \tag{7}$$

Consider that $\hat{y}_k$ is assume to be one or zero, we can obtain the following solution:

$$\hat{y}_k = \begin{cases} 1 & \text{if } f_k(\boldsymbol{x}) \geq \tau(\alpha_k), \\ 0 & \text{if } f_k(\boldsymbol{x}) \leq \tau(\beta_k), \\ -1 & \text{otherwise}, \end{cases} \tag{8}$$

where $\tau(\alpha_k) = \exp(-\alpha_k)$ and $\tau(\beta_k) = 1 - \exp(-\beta_k)$ are two class-aware thresholds, and $\hat{y}_k = -1$ means that the label $\hat{y}_k$ would not be used for model training.

### 3.3 Class-Distribution-Aware Thresholding

An important problem is how to set the thresholds $\tau(\alpha_k)$ and $\tau(\beta_k)$, which determine the numbers of positive and negative pseudo-labels for every class $k$. In order to capture the true class distribution, we propose the Class-distribution-Aware Thresholding (CAT) strategy to determine $\tau(\alpha_k)$ and $\tau(\beta_k)$. Suppose that we are given $\boldsymbol{y}_j, \forall j \in [m]$, *i.e.*, the true label vectors of unlabeled training instances. By solving the following equation, we can obtain $\tau(\alpha_k)$ and $\tau(\beta_k)$ that capture the true class distribution of unlabeled data.

$$\frac{\sum_{j=1}^{m} \mathbb{I}(f_k(\boldsymbol{x}_j) \geq \tau(\alpha_k))}{m} = \gamma_k^*, \quad \frac{\sum_{j=1}^{m} \mathbb{I}(f_k(\boldsymbol{x}_j) \leq \tau(\beta_k))}{m} = \rho_k^*,$$

where $\gamma_k^* = \frac{\sum_{j=1}^{m} \mathbb{I}(y_{jk}=1)}{m}$ and $\rho_k^* = \frac{\sum_{j=1}^{m} \mathbb{I}(y_{jk}=0)}{m}$ are respectively the proportions of positive and negative labels in unlabeled data for class $k$. Although during the training process, the true labels of unlabeled instances are inaccessible, our observation shows that the estimated class distribution, *i.e.*,

the class proportions of positive and negative labels in labeled examples, can tightly approximate the true class distribution. As shown Figure 1 (b), we illustrate the proportions of positive labels in labeled examples and unlabeled examples for every class $k$ on two benchmark datasets COCO and VOC. The proportions of labeled examples are respectively $p = 0.05$ and $p = 0.1$ for COCO and VOC. From the figures, it can be observed that even with a small proportion of labeled examples ($p = 0.05$), it achieves a nearly complete overlap between the estimated and true curves, which validates that the estimated class distribution can be a tight approximation of the true one. This motivate us to alternatively utilize the estimated class distribution to solve the solutions for $\tau(\alpha_k)$ and $\tau(\beta_k)$:

$$\frac{\sum_{j=1}^{m} \mathbb{I}(f_k(\boldsymbol{x}_j) \geq \tau(\alpha_k))}{m} = \hat{\gamma}_k, \quad \frac{\sum_{j=1}^{m} \mathbb{I}(f_k(\boldsymbol{x}_j) \leq \tau(\beta_k))}{m} = \hat{\rho}_k, \tag{9}$$

where $\hat{\gamma}_k = \frac{\sum_{i=1}^{n} \mathbb{I}(y_{ik}=1)}{n}$ and $\hat{\rho}_k = \frac{\sum_{i=1}^{n} \mathbb{I}(y_{ik}=0)}{n}$ are respectively the proportions of positive and negative labels in labeled data for class $k$. Figure 1 provides an illustration of the comparison between three instance-aware pseudo-labeling methods and the CAP method. By utilizing CAT strategy, CAP is expected to assign pseudo-labels with the class distribution that approximates the true one.

In practice, to further improve the performance of CAP, one feasible solution is to discard a fraction of unreliable pseudo-labels with relatively low confidences, which may have a negative impact on the model training. Specifically, for any class $k \in [q]$, we select top $\eta_1 \cdot \hat{\gamma}_k$ and $\eta_0 \cdot \hat{\rho}_k$ proportion probable pseudo-labels, where $\eta_1, \eta_0 \in [0, 1]$ are two parameters to control the reliable intervals of pseudo-labels. By substituting the two terms into the right sides of Eq.(9), we can obtain the thresholds correspondingly. In Section 5.4, we perform ablation experiments to study the influence of reliable intervals on the model performance.

## 4 Theoretical Analysis

In this section, we perform theoretical analyses for the proposed method. In general, the performance of pseudo-labeling depends mainly on two factors, i.e., the quality of the model predictions and the correctness of estimated class distribution. Our work focuses on the latter. Consider an extreme case, where the model predictions are perfect, i.e., the confidences of positive labels are always greater than that of negative labels. In such a case, we still need an appropriate threshold to precisely separate the positive and negative labels. This implies that we need to capture the true class distribution of unlabeled data in order to achieve desirable pseudo-labeling performance.

### 4.1 Correctness of the Estimated Class Distribution

To study the correctness of estimated class distribution, we provide the following theorem, which gives an upper bound on the difference between the estimated class proportion $\hat{\gamma}_k$ and the true class proportion $\gamma_k^*$ (its proof is given in Appendix A). A similar result can be derived for $\hat{\rho}_k$.

**Theorem 1.** *Assume the estimated class proportion $\hat{\gamma}_k = \frac{1}{n} \sum_{i=1}^{n} \mathbb{I}(y_{ik} = 1)$, and the true class proportion $\gamma_k^* = \frac{1}{m} \sum_{j=1}^{m} \mathbb{I}(y_{jk} = 1)$ for any $k \in [q]$, where $n$ and $m$ are the numbers of labeled and unlabeled examples that satisfy $m >> n$. Then, with the probability larger than $1 - 2n^{-1} - 2m^{-1}$, we have, $\forall k \in [q], |\hat{\gamma}_k - \gamma_k^*| \leq \frac{\sqrt{\log n}}{\sqrt{2n}} + \frac{\sqrt{\log m}}{\sqrt{2m}}$.*

Theorem 1 tells us that the correctness of the estimated class distribution mainly depends on the number of labeled and unlabeled data. In general, the bound is dominated by the first term, since it always satisfies $m >> n$. By neglecting the second term, we can see that $\forall k \in [q], \gamma_k^* \to \hat{\gamma}_k$ in the parametric rate $\mathcal{O}_p(1/\sqrt{n})$, where $\mathcal{O}_p$ denotes the order in probability. Obviously, as the number of training examples increase, the estimated class distribution would quickly converge to the true one.

### 4.2 Generalization Bound

Moreover, we study the generalization performance of CAP. Before providing the main results, we first define the true risk with respect to the classification model $f(\boldsymbol{x}; \theta)$:

$$R(f) = \mathbb{E}_{(\boldsymbol{x}, \boldsymbol{y})} \left[ \mathcal{L}(f(\boldsymbol{x}), \boldsymbol{y}) \right].$$

Our goal is to learn a good classification model by minimizing the empirical risk $\widehat{R}(f) = \widehat{R}_l(f) + \widehat{R}_u(f)$, where $\widehat{R}_l(f)$ and $\widehat{R}_u(f)$ are respectively the empirical risk of the labeled loss $\mathcal{L}_l(f(\boldsymbol{x}), \boldsymbol{y})$ and unlabeled loss $\mathcal{L}_u(f(\boldsymbol{x}), \boldsymbol{y})$:

$$\widehat{R}_l(f) = \frac{1}{n} \sum_{i=1}^{n} \mathcal{L}(f(\boldsymbol{x}_i), \boldsymbol{y}_i), \quad \widehat{R}_u(f) = \frac{1}{m} \sum_{j=1}^{m} \mathcal{L}_u(f(\boldsymbol{x}_j), \boldsymbol{y}_j).$$

Note that during the training, we cannot train a model directly by optimizing $\widehat{R}_u(f)$, since the labels of unlabeled data are inaccessible. Instead, we train the model with $\widehat{R}'_u(f) = \frac{1}{m} \sum_{j=1}^{m} \mathcal{L}_u(f(\boldsymbol{x}_j), \hat{\boldsymbol{y}}_j)$, where $\hat{\boldsymbol{y}}_j$ represents the pseudo-label vector of the instance $\boldsymbol{x}_j$.

Let $\ell(f_k(\boldsymbol{x})) = y_k \ell_1(f_k(\boldsymbol{x})) + (1 - y_k)\ell_0(f_k(\boldsymbol{x}))$ be the loss for the class $k$, and $L_\ell$ be any (not necessarily the best) Lipschitz constant of $\ell$. Let $\mathcal{R}_N(\mathcal{F})$ be the expected Rademacher complexity [31] of $\mathcal{F}$ with $N = m + n$ training points. Let $\hat{f} = \arg\min_{f \in \mathcal{F}} \widehat{R}(f)$ be the empirical risk minimizer, where $\mathcal{F}$ is a function class, and $f^\star = \arg\min_{f \in \mathcal{F}} R(f)$ be the true minimizer. We derive the following theorem, which provides a generalization error bound for the proposed method (its proof is given in Appendix B).

**Theorem 2.** *Suppose that $\ell(\cdot)$ is bounded by $B$. For some $\epsilon > 0$, if $\sum_{j=1}^{m} |\mathbb{I}(f_k(x_j) \geq \tau(\alpha_k)) - \mathbb{I}(y_{jk} = 1)|/m \leq \epsilon$ for any $k \in [q]$, for any $\delta > 0$, with probability at least $1 - \delta$, we have*

$$R(\hat{f}) - R(f^\star) \leq 2qB\epsilon + 4qL_\ell \mathcal{R}_N(\mathcal{F}) + 2qB\sqrt{\frac{\log \frac{2}{\delta}}{2N}}.$$

From Theorem 4, it can be observed that the generalization performance of $\hat{f}$ mainly depends on two factors, *i.e.*, the pseudo-labeling error $\epsilon$ and the number of training examples $N$. Apparently, a smaller pseudo-labeling error $\epsilon$ often leads to better generalization performance. Thanks to its ability to capture the true class distribution, CAP can achieve a much smaller pseudo-labeling error $\epsilon$ than existing instance-aware pseudo-labeling methods, which is beneficial for obtaining better classification performance. This can be further validated by our empirical results in Section 5.3. The second factor is the number of training examples. As $N \to \infty$ and $\epsilon \to 0$, Theorem 4 shows that the empirical risk minimizer $\hat{f}$ converges to the true risk minimizer $f^\star$.

## 5 Experiments

In this section, we first perform experiments to validate the effectiveness of the proposed method; then, we perform ablation studies to analyze the mechanism behind CAP.

### 5.1 Experimental Settings

**Datasets** To evaluate the performance of the propose method, we conduct experiments on three benchmark image datasets, including Pascal VOC-2012 (VOC for short) [2] [12], MS-COCO-2014 (MS-COCO for short) [3] [25], and NUS-WIDE (NUS for short) [4] [7]. The detailed information of these datasets can be found in the appendix. For each dataset, we randomly sample a proportion $p \in \{0.05, 0.1, 0.15, 0.2\}$ of examples with full labels while the others without any supervised information. Following the previous works [8, 53], we report the mean average precision (mAP) on the test set for each method.

**Comparing methods** To validate the effectiveness of the proposed method, we compare it with five groups of methods: 1) three instance-aware pseudo-labeling methods: **Top-1** (Eq.(2)), **Top-$k$** (Eq.(3)), **IAT** (Eq.(4)); 2) two state-of-the-art MLML methods: **LL** [19] (includes three variants **LL-R**, **LL-Ct**, and **LL-Cp**), **PLC** [46]; 3) Two state-of-the-art SSL methods: **Adsh** [14], **FreeMatch** [44]; 4) One state-of-the-art SSMLL method: **DRML** [43]; 5) Two baseline methods, **BCE**, **ASL** [34]. DRML is the only deep SSMLL method whose source code could be found on the Internet.

---

[2] http://host.robots.ox.ac.uk/pascal/VOC/

[3] https://cocodataset.org

[4] https://lms.comp.nus.edu.sg/wp-content/uploads/2019/research/nuswide/NUS-WIDE.html

Table 1: Comparison results on VOC and COCO in terms of mAP (%). The best performance is highlighted in bold.

| Method | VOC | | | | COCO | | | |
|---|---|---|---|---|---|---|---|---|
| | $p = 0.05$ | $p = 0.1$ | $p = 0.15$ | $p = 0.2$ | $p = 0.05$ | $p = 0.1$ | $p = 0.15$ | $p = 0.2$ |
| BCE | 67.95 | 75.35 | 78.19 | 79.38 | 58.90 | 63.75 | 65.91 | 67.33 |
| ASL | 71.46 | 78.00 | 79.69 | 80.77 | 59.12 | 63.82 | 66.10 | 67.51 |
| LL-R | 75.69 | 80.96 | 82.31 | 83.55 | 59.31 | 64.25 | 66.61 | 68.01 |
| LL-Ct | 75.77 | 81.04 | 82.31 | 83.50 | 59.33 | 64.23 | 66.69 | 68.11 |
| LL-Cp | 75.79 | 81.03 | 82.36 | 83.68 | 59.27 | 64.19 | 66.68 | 68.12 |
| PLC | 74.49 | 80.35 | 82.35 | 83.39 | 59.85 | 65.03 | 67.62 | 69.14 |
| Top-1 | 75.77 | 80.78 | 82.65 | 83.72 | 57.62 | 62.84 | 65.50 | 66.96 |
| Top-$k$ | 75.07 | 80.20 | 81.99 | 83.16 | 58.25 | 63.52 | 66.11 | 67.49 |
| IAT | 73.24 | 80.27 | 82.39 | 83.55 | 60.34 | 65.54 | 67.88 | 69.25 |
| ADSH | 75.37 | 80.34 | 82.80 | 83.93 | 60.75 | 65.37 | 67.70 | 69.01 |
| FreeMatch | 75.11 | 80.66 | 82.63 | 83.60 | 59.94 | 64.46 | 66.79 | 68.04 |
| DRML | 61.77 | 71.01 | 72.98 | 74.49 | 53.60 | 57.06 | 58.53 | 59.24 |
| Ours | **76.16** | **82.16** | **83.48** | **84.41** | **62.43** | **67.36** | **69.11** | **70.41** |

Table 2: Comparison results on NUS in terms of mAP (%). The best performance is highlighted in bold.

| Method | ASL | LL-R | PLC | Top-1 | Top-$k$ | IAT | ADSH | FreeMatch | DRML | Ours |
|---|---|---|---|---|---|---|---|---|---|---|
| $p = 0.05$ | 42.87 | 40.20 | 43.55 | 40.99 | 40.89 | 42.58 | 43.94 | 43.12 | 30.61 | **44.82** |
| $p = 0.10$ | 46.50 | 44.95 | 47.51 | 45.07 | 45.04 | 46.60 | 47.28 | 46.65 | 35.09 | **48.24** |
| $p = 0.15$ | 48.42 | 47.32 | 49.75 | 47.43 | 47.22 | 48.76 | 49.22 | 48.74 | 37.91 | **49.90** |
| $p = 0.20$ | 49.65 | 48.31 | 50.71 | 48.49 | 48.37 | 49.62 | 49.93 | 49.59 | 39.98 | **51.06** |

Furthermore, most MLML methods cannot be applied to the SSMLL scenario, since they assume that a subset of labels have been annotated for each training instance. The detailed information of these methods can be found in the appendix.

**Implementation**   We employ ResNet-50 [16] pre-trained on ImageNet [36] for training the classification model. We adopt RandAugment [9] and Cutout [10] for data augmentation. We employ AdamW [29] optimizer and one-cycle policy scheduler [10] to train the model with maximal learning rate of 0.0001. The number of warm-up epochs is set as 12 for all datasets. The batch size is set as 32, 64, and 64 for VOC, MS-COCO, and NUS. Furthermore, we perform exponential moving average (EMA) for the model parameter $\theta$ with a decay of 0.9997. For all methods, we use the ASL loss as the base loss function, since it shows superiority to BCE loss [34]. We perform all experiments on GeForce RTX 3090 GPUs. The random seed is set to 1 for all experiments.

## 5.2   Comparison Results

Table 1 and Table 2 report the comparison results between CAP and the comparing methods in terms of mAP on VOC, COCO, and NUS. From the tables, we can see that: 1) DRML obtains unfavorable performance, even worse than baselines BCE and ASL, since it performs two-stage training, which may destroy its representation learning. The original paper did not report the results on these three datasets. Therefore, it is rather important to design an effective SSMLL method in deep learning paradigm. 2) CAP outperforms three instance-aware pseudo-labeling methods, which demonstrates that by utilizing CAT strategy, CAP can precisely estimate the class distribution of unlabeled data and thus obtain desirable pseudo-labeling performance. 3) The performance of CAP is better than that of two state-of-the-art SSL methods. To achieve better performance, we have made several modifications for these two methods not limited to the following: a) use the ASL loss ; b) adopt stronger data augmentations; c) change the training scheme to make them more suitable for the multi-label scenario. 4) CAP achieves the best performance in all cases and significantly outperforms

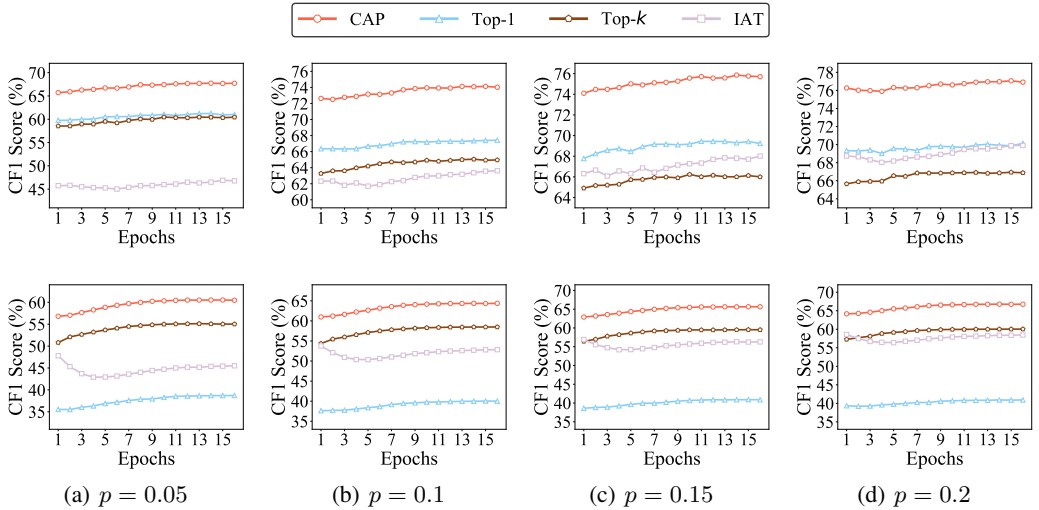

Figure 2: Pseudo-labeling performance in terms of CF1 score on VOC, COCO. Each row corresponds one dataset.

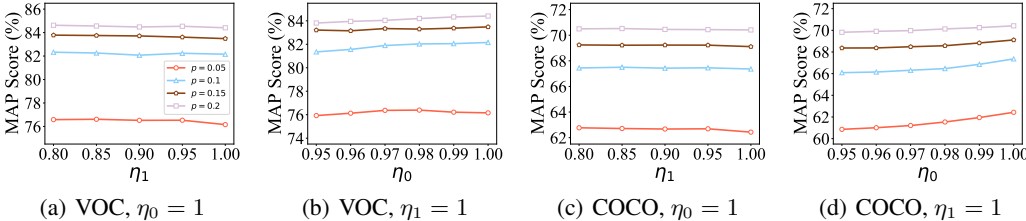

Figure 3: Performance of CAP on VOC and COCO in terms of mAP (%) with the increase of $\eta_1$ and $\eta_0$.

the comparing methods, especially when the number of labeled examples is small. These results convincingly validate the effectiveness of the proposed method.

### 5.3 Study on the Performance of Pseudo-Labeling

In this section, we explain why CAP is better than the conventional instance-aware pseudo-labeling methods. Figure 2 illustrates the performance of different pseudo-labeling methods in terms of CF1 score on VOC and COCO. From the figures, we can see that CAP achieves the best performance in all cases. As discussed above, the pseudo-labeling performance mainly depends on two factors, *i.e.*, the quality of model predictions and the correctness of estimated class proportions. CAP improves the pseudo-labeling performance by precisely estimating the class distribution. An interesting observation is that at the first epoch, when the model predictions are the same for four methods, our method significantly outperforms the comparing methods, since it is able to capture the true class proportions. These results validate that CAP can achieve better pseudo-labeling performance.

### 5.4 Study on the Influence of Reliable Interval

As mentioned above, to improve the performance, instead of using all pseudo-labels, we can train the model with only reliable pseudo-labels within the reliable intervals that are controlled by $\eta_1, \eta_0$. Figure 3 illustrates the performance of CAP as $\eta_1$ and $\eta_0$ change in the ranges of $[0.8, 0.85, 0.9, 0.95, 1]$ and $[0.95, 0.96, 0.97, 0.98, 0.99, 1]$. From the figures, it can be observed that discarding the unreliable positive pseudo-labels would improve the performance, but discarding the unreliable negative pseudo-labels would degrade the performance. One possible reason behinds the phenomenon is due to the significant positive-negative imbalance in MLL data, *i.e.*, the number of negative labels is often much greater than that of positive labels. This leads the model to be sensitive to false positive labels, while be robust to false negative labels. In our main experiments (Table 1 and Table 2), we set

$\eta_1 = 1, \eta_0 = 1$. In practice, we are expected to achieve better performance by tuning the parameter $\eta_1$.

## 6   Conclusion

The paper studies the problem of semi-supervised multi-label learning, which aims to train a multi-label classifier by leveraging the information of unlabeled data. Different from the conventional instance-aware pseudo-labeling methods, we propose to assign pseudo-labels to unlabeled instances in a class-aware manner, with the aim of capturing the true class distribution of unlabeled data. Towards this goal, we propose the CAT strategy to obtain an estimated class distribution, which has been proven to be a desirable estimation of the true class distribution based on our observations. Theoretically, we first perform an analysis on the correctness of estimated class distribution; then, we provide the generalization error bound for CAP and show its dependence to the pseudo-labeling performance. Extensive experimental results on multiple benchmark datasets validate that CAP can achieve state-of-the-art performance.

In general, the performance of pseudo-labeling depends mainly on two factors, i.e., the quality of the model predictions and the correctness of the estimated class distribution. This work focuses on the latter. In future, we plan to boost the performance of SSMLL by improving the quality of model predictions.

## Acknowledgments and Disclosure of Funding

Sheng-Jun Huang was supported by Natural Science Foundation of Jiangsu Province of China (BK20222012, BK20211517), the National Key R&D Program of China (2020AAA0107000), and NSFC (62222605). Masashi Sugiyama was supported by JST CREST Grant Number JPMJCR18A2.

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

## A    Proof of Theorem 1

**Theorem 3.** *Assume the estimated class proportion $\hat{\gamma}_k = \frac{1}{n}\sum_{i=1}^{n}\mathbb{I}(y_{ik}=1)$, and the true class proportion $\gamma_k^* = \frac{1}{m}\sum_{j=1}^{m}\mathbb{I}(y_{jk}=1)$ for any $k \in [q]$, where $n$ and $m$ are the numbers of labeled and unlabeled examples that satisfy $m >> n$. Then, with the probability larger than $1 - 2n^{-1} - 2m^{-1}$, we have, $\forall k \in [q], |\hat{\gamma}_k - \gamma_k^*| \le \frac{\sqrt{\log n}}{\sqrt{2n}} + \frac{\sqrt{\log m}}{\sqrt{2m}}$.*

*Proof.* The proof is mainly based on Hoeffding's inequality that can be defined as follows.

**Lemma 1.** *(Hoeffding's inequality). Let $z_1, ..., z_N$ be independent random variables bounded by $[a_i, b_i]$. Then $\hat{z} = \frac{1}{N}\sum_{i=1}^{N}z_i$ obeys for any $\nu > 0$*

$$\Pr(|\hat{z} - \mathbb{E}[\hat{z}] \ge \nu|) \le 2\exp(-\frac{2N^2\nu^2}{\sum_{i=1}^{N}(b_i - a_i)^2}).$$

Let $\bar{\gamma}_k = p(y_k = 1)$ represents the expected class proportion. According to Hoeffding's inequality, for any $k \in [q]$, we have

$$\Pr(|\hat{\gamma}_k - \bar{\gamma}_k| \le \frac{\sqrt{\log n}}{\sqrt{2n}}) \ge 1 - 2n^{-1}$$

or equivalently, with the probability at least $1 - 2n^{-1}$, we have $|\hat{\gamma}_k - \bar{\gamma}_k| \le \sqrt{\log n}/\sqrt{2n}$. Similarly, with the probability at least $1 - 2m^{-1}$, for any $k \in [q]$, we have $|\gamma_k^* - \bar{\gamma}_k| \le \sqrt{\log m}/\sqrt{2m}$. By applying the triangle inequality, with the probability at least $1 - 2n^{-1} - 2m^{-1}$, for any $k \in [q]$, we have

$$|\gamma_k^* - \hat{\gamma}_k| \le \frac{\sqrt{\log n}}{\sqrt{2n}} + \frac{\sqrt{\log m}}{\sqrt{2m}}.$$

which completes the proof. $\qquad\square$

## B    Proof of Theorem 2

**Theorem 4.** *Suppose that $\ell(\cdot)$ is bounded by $B$. For some $\epsilon > 0$, if $\sum_{j=1}^{m}|\mathbb{I}(f_k(x_j) \ge \tau(\alpha_k)) - \mathbb{I}(y_{jk}=1)|/m \le \epsilon$ for any $k \in [q]$, for any $\delta > 0$, with probability at least $1 - \delta$, we have*

$$R(\hat{f}) - R(f^*) \le 2qB\epsilon + 4qL_\ell R_N(\mathcal{F}) + 2qB\sqrt{\frac{\log\frac{2}{\delta}}{2N}}.$$

*Proof.* Before proving the theorem, we first provide two useful lemmas as follows.

We primarily derive the uniform deviation bound between $R(f)$ and $\widehat{R}(f)$, which is a simple extension of the result in the binary setting [31].

**Lemma 2.** *Suppose that the loss function $\ell$ is $L_\ell$-Lipschitz continuous w.r.t. $\theta$. For any $\delta > 0$, with probability at least $1 - \delta$, we have*

$$|R(f) - \widehat{R}(f)| \le 2qL_\ell \mathcal{R}_{n+m}(\mathcal{F}) + qB\sqrt{\frac{\log\frac{2}{\delta}}{2(n+m)}} \qquad (1)$$

*Proof.* In order to prove this lemma, we define the Rademacher complexity of $\mathcal{L}$ and $\mathcal{F}$ with $m + n$ training examples as follows:

$$\mathcal{R}_{n+m}(\mathcal{L} \circ \mathcal{F})$$

$$= \mathbb{E}_{\boldsymbol{x},\boldsymbol{y},\boldsymbol{\sigma}}\left[\sup_{f\in\mathcal{F}}\sum_{i=1}^{n}\sigma_i\mathcal{L}(f(\boldsymbol{x}_i),\boldsymbol{y}_i) + \sum_{j=1}^{m}\sigma_j\mathcal{L}(f(\boldsymbol{x}_j),\boldsymbol{y}_j)\right]$$

Considering that $\mathcal{L}(f(\boldsymbol{x}), \boldsymbol{y}) = \sum_{k=1}^{q} \ell(f_k, y_k)$, we have

$$\begin{aligned}
\mathcal{R}_{n+m}(\mathcal{L} \circ \mathcal{F}) &\leq q\mathcal{R}_{n+m}(\ell \circ \mathcal{F}) \\
&\leq qL_\ell \mathcal{R}_{n+m}(\mathcal{F})
\end{aligned} \tag{10}$$

where the second line is due to the Lipschitz continuity of the loss function $\ell$.

Then, we proceed the proof by showing that the one direction $\sup_{f \in \mathcal{F}} R(f) - \widehat{R}(f)$ is bounded with probability at least $1 - \delta/2$, and the other direction can be proved similarly. Note that replacing an example $(\boldsymbol{x}_j, \boldsymbol{y}_j)$ with another $(\boldsymbol{x}_j', \boldsymbol{y}_j')$ leads to a change of $\sup_{f \in \mathcal{F}} R(f) - \widehat{R}(f)$ at most $\frac{qB}{n+m}$ due to the fact that $\ell$ is bounded by $B$. According to *McDiarmid's inequality* [31], for any $\delta > 0$, with probability at least $1 - \delta/2$, we have

$$\sup_{f \in \mathcal{F}} R(f) - \widehat{R}(f) \leq \mathbb{E}\left[\sup_{f \in \mathcal{F}} R(f) - \widehat{R}(f)\right] + qB\sqrt{\frac{\log \frac{2}{\delta}}{2(n+m)}} \tag{11}$$

According to the result in [31] (Theorem 3.3) that shows $\mathbb{E}[\sup_{f \in \mathcal{F}} R(f) - \widehat{R}(f)] \leq 2\mathcal{R}_m(\mathcal{F})$, by further considering the other direction $\sup_{f \in \mathcal{F}} \widehat{R}(f) - R(f)$, with probability at least $1 - \delta$, we have

$$\sup_{f \in \mathcal{F}} \left| R(f) - \widehat{R}(f) \right| \leq 2qL_\ell \mathcal{R}_m(\mathcal{F}) + qB\sqrt{\frac{\log \frac{2}{\delta}}{2n+m}} \tag{12}$$

which completes the proof. $\qquad \square$

Then, we can bound the difference between $\widehat{R}(f)$ and $\widehat{R}'(f)$ as follows

**Lemma 3.** *Suppose that $\ell(\cdot)$ is bounded by $B$. For some $\epsilon > 0$, if $\sum_{j=1}^{m} |\mathbb{I}(f_k(x_j) \geq \tau(\alpha_k)) - \mathbb{I}(y_{jk} = 1)|/m \leq \epsilon$ for any $k \in [q]$, for any $f \in \mathcal{F}$, we have:*

$$\left| \widehat{R}_u'(f) - \widehat{R}_u(f) \right| \leq qB\epsilon \tag{13}$$

*Proof.* Without loss of generality, assume that $\epsilon$ is the largest pseudo-labeling error among $q$ classes, *i.e.*, $\epsilon = \max_{k \in [q]} \sum_{j=1}^{m} |\mathbb{I}(f_k(x_j) \geq \tau(\alpha_k)) - \mathbb{I}(y_{jk} = 1)|/m$. Obviously, $\epsilon$ consists of exactly two types of pseudo-labeling error:

$$\begin{aligned}
\epsilon_1 &= \frac{\sum_{j=1}^{m} \mathbb{I}(f_k(\boldsymbol{x}_j) < \tau(\alpha_k), y_{jk} = 1)}{m} \\
\epsilon_0 &= \frac{\sum_{j=1}^{m} \mathbb{I}(f_k(\boldsymbol{x}_j) \geq \tau(\alpha_k), y_{jk} = 0)}{m}
\end{aligned} \tag{14}$$

where $\epsilon_1$ calculates the proportion of positive labels being treated as negative ones, and $\epsilon_0$ calculates the proportion of negative labels being treated as positive ones. Then, we prove the following two sides, which provide the bounds for $\widehat{R}_u'(f)$. Firstly, we prove its upper bound:

$$\begin{aligned}
\widehat{R}_u'(f) &= \frac{1}{m} \sum_{j=1}^{m} \sum_{k=1}^{q} \mathbb{I}(f_k(\boldsymbol{x}_j) \geq \tau(\alpha_k))\ell_1(f_k(\boldsymbol{x}_j)) + \mathbb{I}(f_k(\boldsymbol{x}_j) < \tau(\alpha_k))\ell_0(f_k(\boldsymbol{x}_j)) \\
&\leq \frac{1}{m} \sum_{j=1}^{m} \sum_{k=1}^{q} \mathbb{I}(y_{jk} = 1)\ell_1(f_k(\boldsymbol{x}_j)) + \mathbb{I}(y_{jk} = 0)\ell_0(f_k(\boldsymbol{x}_j)) \\
&\qquad + \mathbb{I}(y_{jk} = 0, f_k(\boldsymbol{x}_j) \geq \tau(\alpha_k))\ell_1(f_k(\boldsymbol{x}_j)) + \mathbb{I}(y_{jk} = 1, f_k(\boldsymbol{x}_j) < \tau(\alpha_k))\ell_0(f_k(\boldsymbol{x}_j)) \\
&\leq \frac{1}{m} \sum_{j=1}^{m} \mathcal{L}(f(\boldsymbol{x}_j), \boldsymbol{y}_j) + \epsilon_0 \sum_{k=1}^{q} \ell_1(f_k(\boldsymbol{x}_j)) + \epsilon_1 \sum_{k=1}^{q} \ell_0(f_k(\boldsymbol{x}_j)) \\
&\leq \widehat{R}_u(f) + qB\epsilon
\end{aligned}$$

where the second line holds based on Eq.(14). Then, we prove its low bound:

$$\widehat{R}'_u(f) = \frac{1}{m}\sum_{j=1}^{m}\sum_{k=1}^{q}\mathbb{I}(f_k(\boldsymbol{x}_j) \geq \tau(\alpha_k))\ell_1(f_k(\boldsymbol{x}_j)) + \mathbb{I}(f_k(\boldsymbol{x}_j) < \tau(\alpha_k))\ell_0(f_k(\boldsymbol{x}_j))$$

$$\geq \frac{1}{m}\sum_{j=1}^{m}\sum_{k=1}^{q}\mathbb{I}(y_{jk}=1)\ell_1(f_k(\boldsymbol{x}_j)) + \mathbb{I}(y_{jk}=0)\ell_0(f_k(\boldsymbol{x}_j))$$

$$\qquad - \mathbb{I}(y_{jk}=1, f_k(\boldsymbol{x}_j) < \tau(\alpha_k))\ell_1(f_k(\boldsymbol{x}_j)) - \mathbb{I}(y_{jk}=0, f_k(\boldsymbol{x}_j) \geq \tau(\alpha_k))\ell_0(f_k(\boldsymbol{x}_j))$$

$$\geq \frac{1}{m}\sum_{j=1}^{m}\mathcal{L}(f(\boldsymbol{x}_j), \boldsymbol{y}_j) - \epsilon_1\sum_{k=1}^{q}\ell_1(f_k(\boldsymbol{x}_j)) - \epsilon_0\sum_{k=1}^{q}\ell_0(f_k(\boldsymbol{x}_j))$$

$$\geq \widehat{R}_u(f) - qB\epsilon$$

By combining these two sides, we can obtain the following result:

$$\left|\widehat{R}'_u(f) - \widehat{R}_u(f)\right| \leq qB\epsilon \tag{15}$$

which concludes the proof. $\qquad\square$

For any $\delta > 0$, with probability at least $1 - \delta$, we have:

$$R(\hat{f})$$

$$\leq \widehat{R}_l(\hat{f}) + \widehat{R}_u(\hat{f}) + 2qL_\ell\mathcal{R}_{n+m}(\mathcal{F}) + qB\sqrt{\frac{\log\frac{2}{\delta}}{2N}}$$

$$\leq \widehat{R}_l(\hat{f}) + \widehat{R}'_u(\hat{f}) + qB\epsilon + 2qL_\ell\mathcal{R}_N(\mathcal{F}) + qB\sqrt{\frac{\log\frac{2}{\delta}}{2N}}$$

$$\leq \widehat{R}_l(f) + \widehat{R}'_u(f) + qB\epsilon + 2qL_\ell\mathcal{R}_N(\mathcal{F}) + qB\sqrt{\frac{\log\frac{2}{\delta}}{2N}} \tag{16}$$

$$\leq \widehat{R}_l(f) + \widehat{R}_u(f) + 2qB\epsilon + 2qL_\ell\mathcal{R}_N(\mathcal{F}) + qB\sqrt{\frac{\log\frac{2}{\delta}}{2N}}$$

$$\leq R(f) + 2qB\epsilon + 4qL_\ell\mathcal{R}_N(\mathcal{F}) + 2qB\sqrt{\frac{\log\frac{2}{\delta}}{2N}}$$

where the first and fifth lines are based on Eq.(1), and second and fourth lines are due to Lemma 2. The third line is by the definition of $\hat{f}$. $\qquad\square$

## C  Details of Experimental Settings

Table 3 reports the detailed characteristics of four benchmark datasets. VOC is a popular multi-label dataset that has been divided into a *trainval* set containing 5,011 examples and a test set containing 4,952 examples from 20 object categories. MS-COCO is another widely used multi-label dataset, which consists of 82,081 training examples and 40,504 validation examples belonging to 80 different categories. In our experiments, the validation set is used for testing. NUS is incomplete online due to many invalid URLs. Our collection consists of 126,034 training images and 84,226 testing images from 81 classes.

**Comparing methods** To validate the effectiveness of the proposed method, we compare it with four groups of methods: 1) Three instance-aware pseudo labeling methods: **Top-1**, which is similar to

Table 3: The detailed characteristics of benchmark datasets.

| Dataset | # Training | # Testing | # Classes |
|---------|-----------|-----------|-----------|
| VOC     | 5,011     | 4,952     | 20        |
| COCO    | 82,081    | 16,416    | 80        |
| NUS     | 126,034   | 84,226    | 81        |

FixMatch [38] that selects the most probable label as the ground-truth one; **Top-$k$**, which selects the top $k$ rank labels according to the predicted probabilities as the true labels; **IAT**, which utilizes the instance-aware thresholding strategy to separates positive and negative labels. 2) Two state-of-the-art MLML methods: **LL** [19] (includes three variants **LL-R**, **LL-Ct**, and **LL-Cp**), which rejects or corrects the large-loss examples to prevent model from memorizing noisy labels; **PLC** [46], which performs the pseudo-labeling consistency regularization for recovering the labeling information of potential labels. 3) Two state-of-the-art SSL methods: **Adsh** [14], which solves the class-imbalance issue in SSL by performing adaptive thresholding; **FreeMatch** [44], which improves the performance of FixMatch by introducing a self-adaptive class fairness regularization. 4) One state-of-the-art SSMLL method: **DRML** [43], which performs pseudo labeling by exploring the feature distribution and the label correlation simultaneously. 5) Two baseline methods, **BCE**, which is the most commonly used loss function for multi-label classification; **ASL** [34], which improves BCE by utilizing the dynamic down-weighting and hard-thresholding techniques. DRML is the only deep SSMLL method whose source code could be found on the Internet. Furthermore, most MLML methods cannot be applied to the SSMLL scenario, since they assume that a subset of labels have been annotated for each training instance.

