# OpenReview forum: "Class-Distribution-Aware Pseudo-Labeling for Semi-Supervised Multi-Label Learning"
_NeurIPS.cc/2023/Conference — NeurIPS 2023 poster_

### Official Review · Reviewer_wYDh · 2023-06-28

**Soundness:** 4 excellent
**Presentation:** 4 excellent
**Contribution:** 4 excellent
**Rating:** 7
**Confidence:** 5

**Summary:**

The paper proposes a class-distribution-aware method to deal with the semi-supervised multi-label learning (SSMLL) problem. The main motivation is that the conventional pseudo-labeling methods cannot be applied to SSMLL scenarios, since instance is assigned with more than one ground-truth labels. To solve this challenge, the paper proposes a class-aware pseudo-labeling strategy, which is free of estimating the number of true labels for each instance. To capture the class distribution of unlabeled data, authors propose a regularized learning framework to determine the pseudo-labels by exploiting the estimated class distribution of labeled data.

**Strengths:**

The motivation of this paper is clear. Although pseudo-labeling is a commonly used strategy in single-label case, it cannot be applied to multi-label case due to the unknown label count.

The proposed method is intuitive and effective. Authors propose the class-aware pseudo-labeling strategy to cleverly avoid the estimation of the label count for each instance. Consequently, the problem has been transformed to the task of estimating the class distribution of unlabeled data, which can be easily solved by the proposed class-distribution-aware thresholding method.

Theoretical and empirical studies comprehensively verify the effectiveness of the proposed method. Specifically, the paper studies the correctness of the estimated class distributions theoretically and conduct comparisons with sota methods.


**Weaknesses:**

It seems that only one threshold can determine the positive and negative pseudo-labels for each class. It is suggested to explain the reasons why two thresholds are used in the paper.

It seems that the paper did not discuss how to set the parameter of ASL.

There are some language mistakes. The paper is suggested to proofread the paper carefully.


**Questions:**

The paper considers the estimation of class distribution of unlabeled data. Is there any other factor can affect the pseudo-labeling performance?

---

> ### Author Rebuttal · Authors · 2023-08-08
>
> Thanks for your constructive comments. We are glad that you considered our work “well-structured, easy to follow”. We are glad to answer all your questions.
>
> **Q1:** It seems that only one threshold can determine the positive and negative pseudo-labels for each class. It is suggested to explain the reasons why two thresholds are used in the paper.
>
> **A1:** By utilizing two thresholds, our goal is to identify the positive and negative pseudo-labels with high confidences, which guarantees a high reliability of the pseudo-labels. By training on the reliable pseudo-labels, the model would achieve favorable performance.
>
> **Q2:** It seems that the paper did not discuss how to set the parameter of ASL.
>
> **A2:** Thank you for your suggestions. Since the ASL loss function is not our contribution, we followed the settings in the original paper to use the default values for all parameters.
>
> **Q3:** There are some language mistakes. The paper is suggested to proofread the paper carefully.
>
> **A3:** Thank you for you suggestions. We will carefully check the paper and correct the writing mistakes in the revised version.
>
> **Q4:** Is there any other factor can affect the pseudo-labeling performance?
>
> **A4:** In general, the performance of pseudo-labeling depends mainly on two factors, i.e., the quality of the model predictions and the correctness of the estimated class distribution. We will consider improving the quality of the model predictions in the future version.

---

### Official Review · Reviewer_tMT4 · 2023-07-02

**Soundness:** 3 good
**Presentation:** 3 good
**Contribution:** 3 good
**Rating:** 5
**Confidence:** 4

**Summary:**

This work studies pseudo-labeling for multi-label semi-supervised learning. Differing from the traditional instance-aware pseudo-labeling methods, they propose to assign pseudo-labels to unlabeled data in a class-aware manner to capture the true class distribution of the unlabeled data. This work proposes CAT strategy to estimate the class distribution. It proves it is a desirable estimation by performing an analysis of the correctness of the estimation and providing the generalization error bound for CAP.

**Strengths:**

This paper is well-structured and easy to follow. The methods are clearly presented in Section 3. They also provide a theoretical analysis in Section 4 of the proposed method. Solid experiments are conducted to show the effectiveness of the proposed methods. Multi-label classification is an important task, and pseudo-labeling for this scenario will be helpful to this community.

**Weaknesses:**

* The key idea relies on the observation that the class proportions of positive and negative labels in labeled examples can tightly approximate the true class distribution. I am concerned if the unlabeled data have totally different distribution, will the assumption relying on such observation still hold true? Therefore, it would be helpful to show some examples when there is a distribution mismatch.

* The technical novelty seems not that clear. It would be great if the authors can re-state and present the novelty of the propose approach.


**Questions:**

1. Adsh and FreeMatch are designed for traditional SSL, how do the authors implement these two algorithms for the SSMLL setting?
2. This method can be used widely. I would re-consider rating if the authors are willing to make the codebase public.

**Limitations:**

They do not claim/discuss any limitation in their main manuscript.

---

> ### Author Rebuttal · Authors · 2023-08-08
>
> Thanks for your constructive comments. We are glad that you considered our work “well-structured, solid experiments, helpful to the community”. We are glad to answer all your questions.
>
> **Q1:** The key idea relies on the observation that the class proportions of positive and negative labels in labeled examples can tightly approximate the true class distribution.
>
> **A1:** Many semi-supervised learning (SSL) methods have explicitly or implicitly adopted this assumption. For example, the early work [1] presented the expectation regularization, which encourages the the marginal distribution of model predictions on unlabeled data to match the marginal distribution of the ground-truth labels that was estimated during training based on the training examples. Many recent works [2-4] have applied this idea in deep semi-supervised learning. Compared with SSL that has achieved great advances,  semi-supervised multi-label learning (SSMLL) (in the context of deep learning) is still in its nascent period of development. Therefore, our paper focuses on the standard setting where labeled and unlabeled examples follow the same distribution.
>
> According to your suggestion, to show the influence of the label shift on the final performance of our method, following the works [5-6], we draw $P(y)$  from a Dirichlet distribution with concentration $\alpha$. We define the degree of label shift  $\Delta=\sum_{k=1}^q|\hat\gamma_k-\gamma_k^*|$ between the labeled and unlabeled data. By using different values of $\alpha$, the degree of label shift $\Delta$ could be varied. From Table 1, we can see that the performance of CAP degrades slightly as the degree of label shift increases significantly. For example, when $p=0.1$ and $\alpha=1$ (the degree of label shift has increased by ten times), the performance of CAP is still better than the comparing methods in Table 1 in the paper.
>
> We will add a more detailed discussion of this point in the revised version.
>
> Table 1. MAP of CAP under different degrees of label shift. * denotes the result without introducing the label shift.
>
> | $\alpha$  |   \|   |   1   |  10   |  100  |     *     |   \|   | $\alpha$  |   1   |  10   |  100  |     *     |
> | :-------: | :----: | :---: | :---: | :---: | :-------: | :----: | :-------: | :---: | :---: | :---: | :-------: |
> | $\Delta$  | **\|** | 1.345 | 0.904 | 0.940 |   0.136   | **\|** | $\Delta$  | 1.428 | 0.998 | 1.019 |   0.098   |
> | $p$ = 0.1 | **\|** | 66.17 | 66.86 | 66.74 | **67.36** | **\|** | $p$ = 0.2 | 68.86 | 69.38 | 69.31 | **70.41** |
>
> Recently, several methods have been developed specifically to solve the label shift problem [5, 6], where the class marginal distributions of the labeled and unlabeled data are different. This is really an interesting future direction of our work. To solve the label shift  problem in the SSMLL scenario, a straightforward strategy is to estimate the class distribution of the unlabeled data using the well-established method [5, 6], and then apply CAP to obtain pseudo-labels.
>
> [1] Simple, Robust, Scalable Semi-Supervised Learning via Expectation Regularization. ICML’07
>
> [2] ReMixMatch: Semi-Supervised Learning with Distribution Alignment and Augmentation Anchoring. ICLR’20
>
> [3] CoMatch: Semi-supervised Learning with Contrastive Graph Regularization. CVPR’20
>
> [4] AdaMatch: A Unified Approach to Semi-Supervised Learning and Domain Adaptation. ICLR’22
>
> [5] Detecting and Correcting for Label Shift with Black Box Predictors. ICML’18
>
> [6] LTF: A Label Transformation Framework for Correcting Target Shift. ICML’20
>
> **Q2:** The technical novelty seems not that clear. It would be great if the authors can re-state and present the novelty of the propose approach.
>
> **A2:**  Thank you for your suggestion. In SSMLL scenarios, conventional pseudo-labeling methods can hardly work when handling instances associated with multiple labels and an unknown label count. Our paper proposes a Class-Aware Pseudo-labeling (CAP) method to assign pseudo-labels in a class-aware manner, which is free of estimating the label count for each instance. This allows us to transform a hard problem, i.e., estimating the label count for each unlabeled instance, into a much easier problem, i.e., estimating the class distribution of unlabeled data. We further propose a class-distribution-aware thresholding strategy to reliably separate positive and negative pseudo-labels based on the estimated class proportions of labeled data, which tightly approximate the true class proportions.  We will re-organize the Introduction section based on the above discussion to highlight our technical novelty.
>
> **Q3:** Adsh and FreeMatch are designed for traditional SSL, how do the authors implement these two algorithms for the SSMLL setting?
>
> **A3:**  In our experiments, we compared our method with SSMLL methods and MLML (Multi-label Learning with Missing Labels) methods. To further validate the effectiveness of the proposed method, we also compared our method with the SSL methods additionally. As mentioned in line 283-284, to achieve better performance, we made several modifications for SSL methods: 1) use the ASL loss (also used in our method); 2) apply the same data augmentations as our method, including RandAugment and Cutout; 3) change the training strategy to make it more suitable for the multi-label scenario, specifically, employing AdamW optimizer and one-cycle policy scheduler. Furthermore, we also tuned the hyperparameters of these methods to achieve better performance.
>
> **Q4:** This method can be used widely. I would re-consider rating if the authors are willing to make the codebase public.
>
> **A4:**  We have submitted the source code of our method as supplementary material. We promise to make the code publicly available as soon as the paper is accepted.
>
> **Q5:** About the limitation of the paper.
>
> **A5:** Due to character limit, please refer to the beginning of the rebuttal.

---

### Official Review · Reviewer_xWdd · 2023-07-05

**Soundness:** 3 good
**Presentation:** 3 good
**Contribution:** 3 good
**Rating:** 6
**Confidence:** 5

**Summary:**

This paper proposes a Class-Aware Pseudo-labeling (CAP) method to solve semi-supervised multi-label learning (SSMLL) problem by controlling the assignment of positive and negative pseudo-labels for each class through a class-distribution-aware thresholding (CAT) strategy.

**Strengths:**

1. This paper proposes a pseudo-labeling strategy for semi-supervised multi-label learning in a class-aware manner.

2. This paper is easy to read and well organized.

3. The experimental results and the proof of Theorems in this paper seem to be solid.


**Weaknesses:**

1. Lack of some references. This paper is devoted to solving the SSMLL problem. Therefore, the authors should review and compare some SSMLL methods and more SSL methods. For example,
[1] Rizve et al., In defense of pseudo-labeling: An uncertainty-aware pseudo-label selection framework for semi-supervised learning, ICLR, 2021.
[2] Xu et al., Dash: Semi-supervised learning with dynamic thresholding, ICML, 2021.
[3] Zhang et al., Flexmatch: Boosting semi-supervised learning with curriculum pseudo labeling, NeurIPS, 2021.

2. What is the motivation for designing the operation $\tau(\alpha_{k})=exp(-\alpha_{k})$? This does not seem to be mentioned in this paper.

3. It is not enough clear about the Class-Distribution-Aware Thresholding. Also, it is suggested that the author gives a complete methodological illustration about CAP and CAT for each class to facilitate the readers' understanding.

4. The random seed in this paper is set to 1 for all experiments. Following the SSL setting, it is necessary to report the mean value with different random seeds as the final performance.

5. The authors should check if the title of the figure corresponds to the content. For example, Figure 2 is titled OF1 score while the results are CF1 scores. In addition, it is suggested to give a number for each formula.

**Questions:**

Please refer to the Weaknesses.

**Limitations:**

The limitations of this paper should be discussed.

---

> ### Author Rebuttal · Authors · 2023-08-08
>
> Thanks for your great efforts for reviewing our paper. We are glad that you considered our work “easy to read, well-organized, solid”. We are glad to answer all your questions.
>
> **Q1:** Lack of some references. This paper is devoted to solving the SSMLL problem. Therefore, the authors should review and compare some SSMLL methods and more SSL methods.
>
> **A1:** In our paper, we have compared the proposed method with the only SSMLL method found, DRML, and two recent SSL methods, Adsh (2022) and FreeMatch (2023), as well as multiple MLML (Multi-label Learning with Missing Labels) methods. We will add more SSL competitors in the future version. Also, we will improve the Related Works section according to the suggestions and add the mentioned relevant works to the revised version as follows.
>
> The relevant works [1-3] will be added to the third paragraph of the Related Works section:
>
> To improve the reliability of pseudo-labels, an uncertainty-aware pseudo-labeling method proposed in [1] selected reliable pseudo-labels based on the prediction uncertainty.
>
> Unlike FixMatch that selects unlabeled examples with a fixed threshold,  Dash [2] selected unlabeled examples with a dynamic threshold, with the goal of achieving better pseudo-labeling performance.
>
> FlexMatch [3] was proposed to select unlabeled examples for every class according to the current learning status of the model.
>
> [1] In defense of pseudo-labeling: An uncertainty-aware pseudo-label selection framework for semi-supervised learning, ICLR, 2021.
>
> [2] Dash: Semi-supervised learning with dynamic thresholding, ICML, 2021.
>
> [3] Flexmatch: Boosting semi-supervised learning with curriculum pseudo labeling, NeurIPS, 2021.
>
> **Q2:** What is the motivation for designing the operation $\tau(\alpha_k)=\exp(-\alpha_k)$?
>
> **A2:** For solving Eq.(7), according to [1], we obtain the closed-form solution $\hat{y}_k=1$ if $-\log(f_k(x_i))\leq\alpha_k$, that is, $\hat{y}_k=1$ if $f_k\geq\exp(-\alpha_k)$.  For notational simplicity, we denote as $\tau(\alpha_k)=\exp(-\alpha_k)$. We will include the above discussion in the revised version.
>
> [1] Self-Paced Learning for Latent Variable Models. NeurIPS 2010.
>
> **Q3**: It is not enough clear about the Class-Distribution-Aware Thresholding. Also, it is suggested that the author gives a complete methodological illustration about CAP and CAT for each class to facilitate the readers' understanding.
>
> **A3:** Thank you for your suggestion. As illustrated in Figure 1(a), Class-aware Pseudo-labeling (CAP) is a pseudo-labeling framework that differs significantly from conventional pseudo-labeling (Instance-Aware Pseudo-labeling, IAP). IAP performs pseudo-labeling for every instance (row), while CAP performs pseudo-labeling for every class (column). This implies that IAP needs to know in advance the label count for every unlabeled instance,  while CAP only needs to know the class proportions of unlabeled data. For IAP, Figure 1(a) lists three existing pseudo-labeling methods (Top-1, Top-k, instance-aware thresholding). Unfortunately, these methods cannot capture the true label count precisely. For our CAP, we propose the class-distribution-aware thresholding, which aims to use the estimated class proportions of labeled data to approximate the true class proportions of unlabeled data. Figure 1(b) shows that even with a small  number of labeled data (the labeled proportion $p=0.05$), the estimated and true class proportions highly agree with each other. We will add more notes and re-organize the caption of Figure 1 to make the illustration clearer in the revised version.
>
> **Q4:** The random seed in this paper is set to 1 for all experiments.
>
> **A4:** For the sake of easy reproducibility, following most of works in multi-label scenarios like [1-3], we report experimental results with a fixed random seed. Such an experimental setting has been adopted by most of multi-label learning literature. To make a fair comparison, the random seed was set as 1 for every comparing method.
>
> [1] Multi-Label Image Recognition with Graph Convolutional Networks. CVPR'19.
>
> [2] Multi-Label Learning from Single Positive Labels. CVPR'21.
>
> [3] Asymmetric Loss for Multi-Label Classification. ICCV'21.
>
> **Q5:** The authors should check if the title of the figure corresponds to the content. Figure 2 is titled OF1 score while the results are CF1 scores. In addition, it is suggested to give a number for each formula.
>
> **A5:** Thank you for your suggestions. The title was mistakenly wrote as OF1.  We will correct the title and label the formula with numbers in the revised version.
>
> **Q6:** The limitations of this paper should be discussed.
>
> **A6:** In general, the performance of pseudo-labeling depends mainly on two factors, i.e., the quality of the model predictions and the correctness of the estimated class distribution. Our work focuses on the latter. It is a promising future direction to boost the pseudo-labeling performance by improving the quality of the model predictions. A straightforward method is to use dual networks, which can prevent errors from accumulating on a single network, thus achieving better generalization performance. We will include a more detailed discussion in the revised version.

---

> > ### Comment · Reviewer_xWdd · 2023-08-11
> > **Thanks for detailed responds**
> >
> > After the above responds, there remain two major concerns:
> > 1. This paper aims at the semi-supervised multi-label learning. And, all experimental results over one run are reported for comparison. However, I am concerned about the robustness of the proposed method. Following the SSL setting, it is necessary to perform experiments with different random seeds, which has a great effect on randomly dividing training data into labeled and unlabeled set. Besides, UPS [1] is the first to explore the semi-supervised multi-label learning problem. Unfortunately, there is no discussion and comparison with it.
> > [1] In defense of pseudo-labeling: An uncertainty-aware pseudo-label selection framework for semi-supervised learning, ICLR, 2021.
> > 2. According to the solution in Section 3.3, the $\tau(\alpha_k)$ and $\tau(\beta_k)$ are calculated by the proportions of positive and negative labels in labeled data, respectively. However, does this strategy still work for cross-domain unlabeled data in the open world?

---

> > > ### Author Response · Authors · 2023-08-13
> > > **Thanks for your further response!**
> > >
> > > We hope that our response could address your concerns, and we are willing to provide further clarification if necessary.
> > >
> > > **About repeated experiments and the comparison with UPS**
> > >
> > > Table 1 reports the comparison results between CAP and the comparing methods. Due to the time limit, we select two strong competitors from the comparing methods in the original paper. According to your suggestion, we also include the SSL method, UPS, in the comparison. From the table, it can be observed that the performance of our method is significantly better than the comparing methods, especially when the number of labeled data is small. We will include these results and the discussion about UPS in the revised version.
> > >
> > > Table 1. Mean and standard deviation of mAP over 5 runs (random seed = 1, 2, 3, 4, 5).
> > >
> > > |      | VOC          |              |              |              | COCO         |              |              |              |
> > > | ---- | ------------ | ------------ | ------------ | ------------ | ------------ | ------------ | ------------ | ------------ |
> > > |      | p=0.05       | p=0.1        | p=0.15       | p=0.2        | p=0.05       | p=0.1        | p=0.15       | p=0.2        |
> > > | CAP  | 77.15 ± 0.58 | 82.54 ± 0.20 | 83.95 ± 0.24 | 85.04 ± 0.32 | 63.11 ± 0.35 | 67.96 ± 0.32 | 69.92 ± 0.41 | 71.23 ± 0.42 |
> > > | IAT  | 74.78 ± 0.82 | 81.11 ± 0.46 | 82.91 ± 0.30 | 84.38 ± 0.44 | 60.82 ± 0.26 | 66.13 ± 0.30 | 68.50 ± 0.32 | 69.94 ± 0.35 |
> > > | PLC  | 75.61 ± 0.67 | 81.58 ± 0.62 | 83.27 ± 0.47 | 84.48 ± 0.57 | 60.20 ± 0.18 | 65.60 ± 0.29 | 68.32 ± 0.35 | 69.85 ± 0.35 |
> > > | UPS  | 76.25 ± 1.41 | 81.23 ± 0.95 | 82.92 ± 0.67 | 83.71 ± 0.61 | 59.16 ± 0.31 | 64.54 ± 0.23 | 66.87 ± 0.22 | 68.29 ± 0.18 |
> > >
> > >
> > >
> > > **About the label shift problem**
> > >
> > > Many semi-supervised learning (SSL) methods have explicitly or implicitly adopted the assumption that the labeled and unlabeled examples follow the same distribution. For example, the early work [1] presented the expectation regularization, which encourages the the marginal distribution of model predictions on unlabeled data to match the marginal distribution of the ground-truth labels that was estimated during training based on the training examples. Many recent works [2-4] have applied this idea in deep semi-supervised learning. Compared with SSL that has achieved great advances,  semi-supervised multi-label learning (SSMLL) (in the context of deep learning) is still in its nascent period of development. Therefore, our paper focuses on the standard setting where labeled and unlabeled examples follow the same distribution.
> > >
> > > To show the influence of the label shift on the final performance of our method, following the works [5-6], we draw $P(y)$  from a Dirichlet distribution with concentration $\alpha$. We define the degree of label shift  $\Delta=\sum_{k=1}^q|\hat\gamma_k-\gamma_k^*|$ between the labeled and unlabeled data. By using different values of $\alpha$, the degree of label shift $\Delta$ could be varied. From Table 1, we can see that the performance of CAP degrades slightly as the degree of label shift increases significantly. For example, when $p=0.1$ and $\alpha=1$ (the degree of label shift has increased by ten times), the performance of CAP is still better than the comparing methods in Table 1 in the paper.
> > >
> > > We will add a more detailed discussion of the assumption in the revised version.
> > >
> > > Table 1. MAP of CAP under different degrees of label shift on COCO. * denotes the result without introducing the label shift.
> > >
> > > | $\alpha$  |   \|   |   1   |  10   |  100  |     *     |   \|   | $\alpha$  |   1   |  10   |  100  |     *     |
> > > | :-------: | :----: | :---: | :---: | :---: | :-------: | :----: | :-------: | :---: | :---: | :---: | :-------: |
> > > | $\Delta$  | **\|** | 1.345 | 0.904 | 0.940 |   0.136   | **\|** | $\Delta$  | 1.428 | 0.998 | 1.019 |   0.098   |
> > > | $p$ = 0.1 | **\|** | 66.17 | 66.86 | 66.74 | **67.36** | **\|** | $p$ = 0.2 | 68.86 | 69.38 | 69.31 | **70.41** |
> > >
> > > Recently, several methods have been developed specifically to solve the label shift problem [5, 6], where the class marginal distributions of the labeled and unlabeled data are different. This is really an interesting future direction of our work. To solve the label shift  problem in the SSMLL scenario, a straightforward strategy is to estimate the class distribution of the unlabeled data using the well-established method [5, 6], and then apply CAP to obtain pseudo-labels.
> > >
> > > [1] Simple, Robust, Scalable Semi-Supervised Learning via Expectation Regularization. ICML’07
> > >
> > > [2] ReMixMatch: Semi-Supervised Learning with Distribution Alignment and Augmentation Anchoring. ICLR’20
> > >
> > > [3] CoMatch: Semi-supervised Learning with Contrastive Graph Regularization. CVPR’20
> > >
> > > [4] AdaMatch: A Unified Approach to Semi-Supervised Learning and Domain Adaptation. ICLR’22
> > >
> > > [5] Detecting and Correcting for Label Shift with Black Box Predictors. ICML’18
> > >
> > > [6] LTF: A Label Transformation Framework for Correcting Target Shift. ICML’20

---

> > > > ### Comment · Reviewer_xWdd · 2023-08-14
> > > > **Thanks for your reply.**
> > > >
> > > > Thank you for the response to my concerns. I have changed my rating to weak accept based on these new experiments.

---

> > > > > ### Author Response · Authors · 2023-08-16
> > > > > **Thanks for the Valuable Response.**
> > > > >
> > > > > Thank you again for your constructive suggestions! Your suggestions are helpful for us to improve the paper. We will improve our paper according to these suggestions.

---

### Official Review · Reviewer_ekLv · 2023-07-06

**Soundness:** 3 good
**Presentation:** 3 good
**Contribution:** 3 good
**Rating:** 7
**Confidence:** 5

**Summary:**

This papers introduces a novel method called Class-Aware Pseudo-Labeling (CAP) to address the challenges of semi-supervised multi-label learning (SSMLL). Traditional pseudo-labeling methods struggle with instances associated with multiple labels and an unknown label count, often leading to the introduction of false positive labels or the omission of true positive ones. The CAP method overcomes these issues by performing pseudo-labeling in a class-aware manner. It introduces a regularized learning framework that incorporates class-aware thresholds, effectively controlling the assignment of positive and negative pseudo-labels for each class. The paper highlights that even with a small proportion of labeled examples, the estimated class distribution can serve as a reliable approximation. This observation led to the development of a class-distribution-aware thresholding strategy to align the pseudo-label distribution with the true distribution. The paper provides theoretical verification of the correctness of the estimated class distribution and offers a generalization error bound for the proposed method. Extensive experiments on multiple benchmark datasets confirm the effectiveness of the CAP method in addressing the challenges of SSMLL problems.

**Strengths:**

1. The Class-Aware Pseudo-Labeling (CAP) method presents a significant innovation in the field of semi-supervised multi-label learning. Its unique approach of performing pseudo-labeling in a class-aware manner addresses the limitations of traditional pseudo-labeling methods, which often struggle with instances associated with multiple labels and an unknown label count.
2. The authors provide a theoretical verification of the correctness of the estimated class distribution, which is a crucial aspect of the Class-Aware Pseudo-Labeling (CAP) method. This theoretical grounding not only validates the method's approach but also enhances its credibility and reliability. Furthermore, the authors offer a generalization error bound for the proposed method. This is an important contribution as it quantifies the expected performance of the CAP method and provides a measure of its robustness.
3. The performance of the Class-Aware Pseudo-Labeling (CAP) method across all datasets and settings in the experiments is a strength of this paper. The fact that the proposed method consistently achieves optimal results demonstrates its effectiveness and robustness.

**Weaknesses:**

The paper does not provide a clear explanation for the choice of using an exponential transformation in line 167 on the threshold value for each class in the Class-Aware Pseudo-Labeling (CAP) method.

**Questions:**

In line 167 of the paper, the paper mention the use of an exponential transformation on the threshold value for each class in the Class-Aware Pseudo-Labeling (CAP) method. Why not just use a number as the threshold? Could the authors provide more details on the rationale behind this choice?

**Limitations:**

The authors have adequately addressed the limitations.

---

> ### Author Rebuttal · Authors · 2023-08-08
>
> Thanks for your appreciation of our paper. We are glad that you considered our work “novel, a significant innovation, an important contribution”. We are glad to answer all your questions.
>
> **Q1:** The paper does not provide a clear explanation for the choice of using an exponential transformation in line 167 on the threshold value for each class in the Class-Aware Pseudo-Labeling (CAP) method.
>
> **A1:**  Considering the BCE loss ($\ell_1(f_k)=-\log(f_k)$ and $\ell_0(f_k)=-\log(1-f_k)$), for solving Eq. (7), according to [1], we obtain the closed-form solution  $\hat{y}_k=1$ if $-\log(f_k(x_i))\leq\alpha_k$; $(1-\hat{y}_k)=1$, if $-\log(1-f_k(x_i))\leq\beta_k$. With simple computations, we have $\hat{y}_k=1$ if $f_k\geq\exp(-\alpha_k)$; $\hat{y}_k=0$, if $f_k\leq 1-\exp(-\beta_k)$. For notational simplicity, we denote as $\tau(\alpha_k)=\exp(-\alpha_k)$ and $\tau(\beta_k)=1-\exp(-\beta_k)$. In the submitted paper, we mistakenly wrote as $\tau(\beta_k)=\exp(-\beta_k)$. We will correct this writing error in the revised version.
>
> [1] Self-Paced Learning for Latent Variable Models. NeurIPS 2010.

---

> > ### Comment · Reviewer_ekLv · 2023-08-14
> >
> > Thank you for your response. Your clarifications have addressed my concerns.

---

### Author Rebuttal · Authors · 2023-08-09

**About the limitation of the paper** (to reviewers xWdd, tMT4)

In general, the performance of pseudo-labeling depends mainly on two factors, i.e., the quality of the model predictions and the correctness of the estimated class distribution. Our work focuses on the latter. It is a promising future direction to boost the pseudo-labeling performance by improve the quality of the model predictions. A straightforward method is to use dual networks, which can prevent errors from accumulating on a single network, thus achieving better generalization performance. We will include a more detailed discussion in the revised version.

---

### Decision · Program_Chairs · 2023-09-21

**Decision:**

Accept (poster)

**Comment:**

This paper studies the semi-supervised multi-label learning (SSMLL) problem. The authors propose a class-distribution-aware pseudo-labeling method termed CAP. Specifically, CAP adopts class-aware thresholds to improve the approximation of pseudo-labels, thereby improving the generalization effects. Both theoretical and empirical studies are conducted to demonstrate the proposed method.
The reviewers mostly hold positive ratings, except for some incomplete statements. During the discussion period, the authors provided more thorough explanations and results, and a reviewer increased the initial score. Overall, I recommend accepting this paper.